# Neuronal congruency effects in macaque prefrontal cortex

Tao Yao ®[1,2] ✉ & Wim Vanduffel ®[1,2,3,4] ✉

The interplay between task-relevant and task-irrelevant information may induce conflicts that impair behavioral performance, a.k.a. behavioral congruency effects. The neuronal mechanisms underlying behavioral congruency effects, however, are poorly understood. We recorded single unit activity in monkey prefrontal cortex using a task-switching paradigm and discovered a neuronal congruency effect (NCE) that is carried by target and distractor neurons which process target and distractor-related information, respectively. The former neurons provide more signal, the latter less noise in congruent compared to incongruent conditions, resulting in a better target representation. Such NCE is dominated by the level of congruency, and is not determined by the task rules the subjects used, their reaction times (RT), the length of the delay period, nor the response levels of the neurons. We propose that this NCE can explain behavioral congruency effects in general, as well as previous fMRI and EEG results in various conflict paradigms.

Cognitive control is essential for flexible goal-directed behavior allowing us to quickly adapt to continuously changing environments and contexts. A proposed cognitive control mechanism is that, depending on internal goals or external context, subjects allocate attention to task-relevant information while they ignore task-irrelevant information to optimize behavioral performance[1–3]. Sometimes, however, task-irrelevant information has also to be processed to some degree, which can interfere with task (relevant) performance. The cocktail party is a classic situation whereby perception of your interlocutor's voice is typically not dominated by surrounding sounds. Yet in some instances surround information might interfere with your conversation, for example when your name is called by a remote individual, not directly participating in the conversation[4]. In laboratory settings, interference from task-irrelevant information is frequently investigated using 'conflict tasks', including the Stroop, Flanker, Simon and Wisconsin card sorting task, but also pro/anti-saccade, and countermanding tasks[5–11]. In high conflict, or incongruent conditions, errors and reaction times increase compared to low conflict or congruent conditions, a behavioral phenomenon known as the *congruency effect*[12,13]. For example, in incongruent trials of the Stroop task, the meaning of a word interferes with the reporting of its color (e.g. "BLUE" printed in red color), while such interference is absent in the congruent trials (e.g. "RED" printed in red color).

Previous imaging and electrophysiological studies in humans and non-human primates revealed fronto-parietal areas as important players in conflict processing[5,13–28]. However, most of these studies focused on conflict detection and resolution, not on the neuronal mechanisms underlying the behavioral congruency effect: "…what makes an incongruent trial slower or more error-prone…"[29]. We aimed to investigate the neuronal implementation of behavioral congruency effects, i.e., the neuronal congruency effect (NCE), which is the foundation of multiple (sometimes contradictory) theories and models[12,16,30–33]. To this end, we recorded single unit activity in the monkey's frontal eye fields (FEF) while the subjects were performing a task-switching paradigm, using either the spatial location or the color of a cue to indicate a target (see below). Across congruent and incongruent conditions, we controlled the visual input, the motor output, and the allocation of voluntary spatial attention. The FEF contains a retinotopic map of visual saliency, as it integrates the bottom-up driven intrinsic saliency of visual stimuli with top-down signals (e.g., attention, experience, reward expectation, goals, knowledge etc.). The peak activity within the saliency map indicates

[1]Department of Neurosciences, Laboratory of Neuro- and Psychophysiology, KU Leuven Medical School, Leuven, Belgium. [2]Leuven Brain Institute, KU Leuven, Leuven, Belgium. [3]Athinoula A. Martinos Center for Biomedical Imaging, Massachusetts General Hospital, Charlestown, MA, USA. [4]Department of Radiology, Harvard Medical School, Boston, MA, USA. ✉e-mail: taoyao12@hotmail.com; WVanduffel@mgh.harvard.edu

the purported target location in the visual field for further processing[34]. Therefore, if a task-irrelevant feature in an incongruent condition is processed to some degree, it may affect the saliency map through spatial or feature-based processes generated within the FEF, or fed to the FEF from neighboring areas within dorsal lateral pre-frontal cortex[35]. Thus, the FEF is an ideal area to investigate differences in target representation between congruent and incongruent conditions, independent whether the spatial location or a feature (color) of a cue is used[36].

Hypothetically, since two stimuli (target and distractor) were presented in our task, two major neuron populations may affect target selection within the FEF: target neurons (with targets inside their receptive fields (RF)), and distractor neurons (with distractors inside their RFs) (Fig. 1a, bottom). The response of target and distractor neurons, corresponding to 'signal' and 'noise' in the context of target representations, have a positive and negative effect on target representations, respectively (Fig. 1a). Therefore, the signal-to-noise ratio (SNR, signal divided by noise) in the current study reflects how well the target can be distinguished from a distractor by a neuron or an area (e.g., FEF). In other words, it reflects how well the target is represented. Please note that the SNR is not related to net responses, spiking variability or noise correlations. We predict that the SNR of a target representation within FEF (carried by both target and distractor neurons) is lower in incongruent than congruent conditions, which can result in a behavioral congruency effect (Fig. 1a, top). Thus, the larger the difference in SNR between congruent and incongruent trials (Δ SNR in Fig. 1a), the larger the behavioral congruency effect. Moreover, there are at least three possible scenarios to achieve lower SNR in incongruent trials (Fig. 1a, bottom): decreased signal (left), increased noise (middle), and decreased signal & increased noise (right). Our first goal was to determine the activity in target and distractor neurons with our paradigm, and which of the three possibilities may explain lower SNR in the FEF during incongruent versus congruent conditions.

Selective attention is a key component required during all conflict tasks[2,29,37,38]. Selective spatial or featural-based attention can also lead to enhanced and suppressed activity in target and distractor neurons, respectively[39–41]. Hence selective attention can also affect the SNR within FEF. Reaction times (RTs) are sensitive to several cognitive operations. For example, they are faster when subjects are more certain about the location of a target, or when attention is more focused on the target[42–46]. To test whether the NCE is also affected by other cognitive modulations that affect RT, we compared the NCE in trials with fast and slow RTs. If the SNR differences between congruent and incongruent conditions are dominated by other cognitive functions indicated by RT, we expect a negative correlation between RT and SNR (Fig. 1b top). On the other hand, if the SNR is dominated by congruency in our experiment, we expect higher SNR for congruent than incongruent conditions, independent of RT (Fig. 1b bottom). Finally, since we used a relatively large range of delays between color-cue and target dimming (660–1950 ms, Fig. 2), we are also able to investigate the effect of delay duration on the NCE.

## Results

### Behavioral congruency effects with a task-switching paradigm

Two rhesus monkeys (*Macaca mulatta*) were trained to perform a task-switching paradigm (Fig. 2, methods). Monkeys were required to covertly attend to a target and detect its dimming by pressing a button while ignoring distractor stimuli. Target locations were determined by a combination of a task rule-cue (color or spatial rule), and a color-cue (pink or red). A trial was considered congruent when the color and spatial location of the color-cue indicated the same target location under both rules (Supplementary Table 1). Otherwise, the trial is incongruent. A conflict arises during incongruent trials since two stimulus features of the color-cue (i.e., its color and spatial location) indicate different target locations. Yet, only one of them is task-relevant as indicated by the task-rule cue. For example, when a pink color-cue is presented on the left side during color-rule trials, the target will be on the right side. In this case, we can investigate how task-irrelevant cue location interferes with behavioral performance, similarly as in the Simon task. During spatial-rule trials, however, a pink cue on the left would be indicative for a target on the left. In that case, we can investigate how the symbolic meaning of the cue (pink = right in the color rule trials) interferes with behavioral performance, similarly as in a Stroop task.

During 29 recording sessions (12 in Monkey S, 17 in Monkey R), we found significant behavioral congruency effects for both task rules, consistently across left and right target locations (Fig. 3 and Supplementary Tables 2 and 3). The average performance was higher for congruent than incongruent trials (monkey S: 93.8% vs. 80.6%, $t_{11} = 16.07$, $p = 5.5e-9$; monkey R: 95.9% vs. 81.6%, $t_{16} = 13.5$, $p = 3.7e-10$, two-tailed paired t-test, all the reported p values are not corrected in this article, and we always used two-tailed t-test or Wilcoxon Signed Rank test). We also found a small but significant average RT difference between congruent and incongruent conditions (Fig. 3c, d, the last pair of bars, average: monkey S: 349 vs. 353 ms; $t_{11} = 2.85$, $p = 0.016$; monkey R: 365 vs. 377 ms, $t_{16} = 7.1$, $p = 2.5e-6$, two-tailed paired t-test). It is

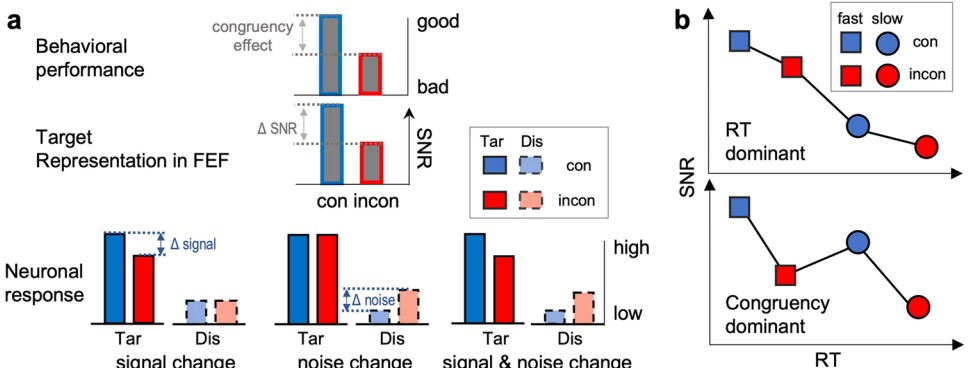

**Fig. 1 | Possible mechanisms underlying behavioral and neuronal congruency effects. a** The top represents the behavioral congruency effect, i.e., worse performance in incongruent than congruent conditions. In brain areas closely related to the output, at least three possible scenarios can lead to a lower signal-to-noise ratio (SNR, middle) and behavioral congruency effects by changing the response amplitudes of target (signal) and/or distractor (noise) neurons. The SNR represents how well a target is represented in FEF (middle). All three possibilities (bottom plots) lead to lower SNR in incongruent conditions. Left: target neurons show lower response in incongruent condition while activity in distractor neurons remains the same (signal change). Middle: distractor neurons show higher response in incongruent condition, while target neurons are the same (noise change). Right: both signal and noise change. **b** Two possible mechanisms explaining SNR in our task. Top: SNR is dominated by RT (indicating other cognitive modulations). Bottom: SNR is dominated by congruency.

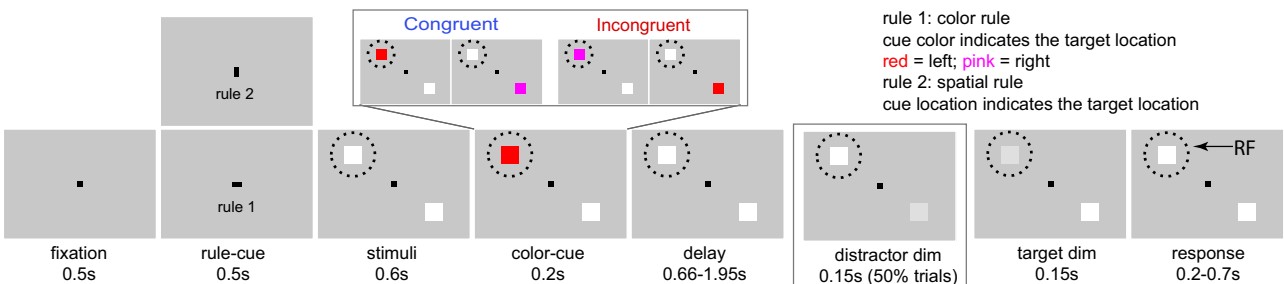

**Fig. 2 | The task paradigm.** Subjects were required to pay covert attention to a target stimulus and respond to its dimming. Subjects initiated trials by foveating a fixation point (FP) which turned to a horizontal or vertical bar (task rule-cue: color or spatial rule) after 500 ms. Then, the FP reappeared simultaneously with two peripheral white squares. Next, one square turned to pink or red, serving as color-cue. In trials with horizontal bars, the color of the color-cue determined target location (red and pink indicating a target on the left and right, respectively), its spatial location being irrelevant. Conversely (vertical bar), the location of the color-cue indicated the position of the target, its color being irrelevant. Trials were subdivided into congruent (i.e., red presented on the left, and pink on right) and incongruent conditions (vice versa) based on the spatial location and color of the color-cue. Monkeys had to respond to target dimming by pressing a button with their left hand to obtain a reward. To ensure that the monkeys paid attention to the target only, monkeys had to ignore distractor dimming, occurring in 50% of the trials (before target dimming). All trial types were pseudo-randomly interleaved. The (virtual) dashed circle indicates the neuron's RF. The monkeys had to foveate to the FP during the entire trial.

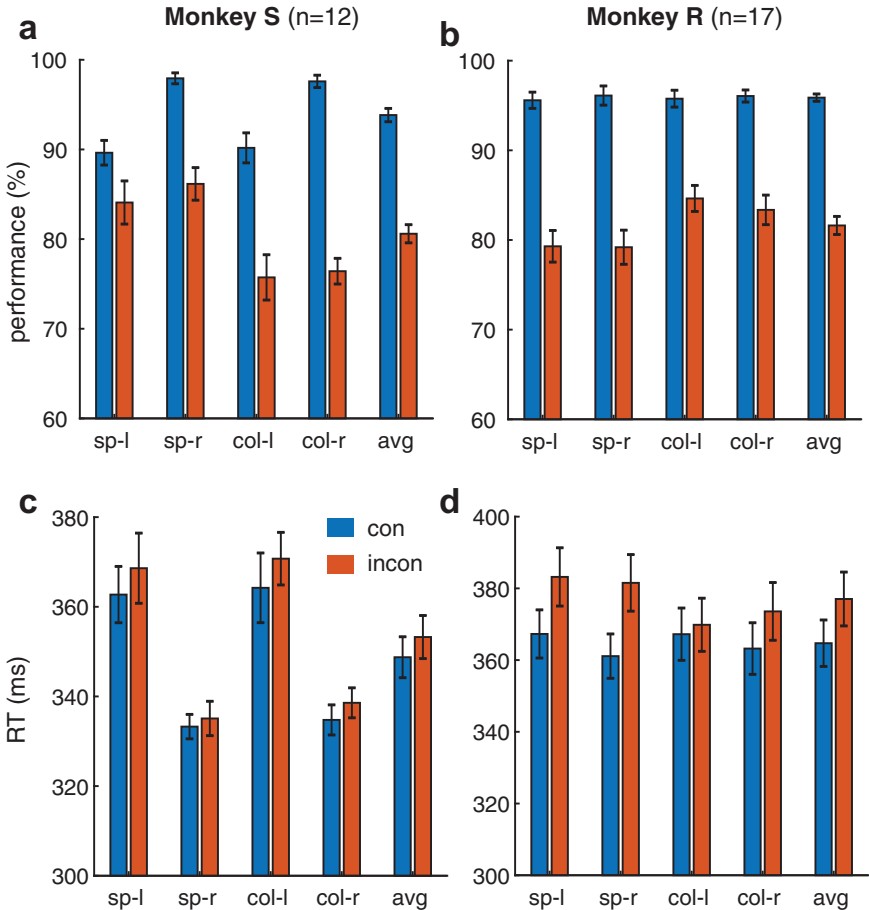

**Fig. 3 | Behavioral congruency effect. a, b** The performance (proportion correctly detected target dimming) for congruent (blue) and incongruent (red) trials during the electrophysiological recording sessions. A behavioral congruency effect was found for both rules and all target locations. All pairwise comparisons between congruent and incongruent conditions were significant (all p < 0.03, two-tailed paired t-test, Supplementary Table 2). **c, d** the average reaction times show a small but significant congruency effect (the last pair of bars in **c**, **d**, p = 0.016 and p = 2.5e-6 for monkey S and R), the other comparison statistics: see Supplementary Table 3. Error bars: SEM across sessions. sp: spatial rule; col: color rule; l: left target; r: right target; avg: average performance across conditions; n = 12 and n = 17 indicates the number of recording sessions for each monkey respectively.

important to note that, unlike traditional conflict tasks, we did not employ a reaction time task. In our study, there was a variable and relatively long gap between the conflict-inducing cue (the color cue) and the go cue (target dimming). Hence monkeys were not forced to make a fast response.

**Congruency modulates both target and distractor neurons**

To study neuronal correlates of the behavioral congruency effect, we recorded single units with 16-channel probes from the right FEF (Supplementary Fig. 1) in two monkeys. In total, 248 visual neurons (121/127 from monkey S/R) with contralateral RFs were included in our

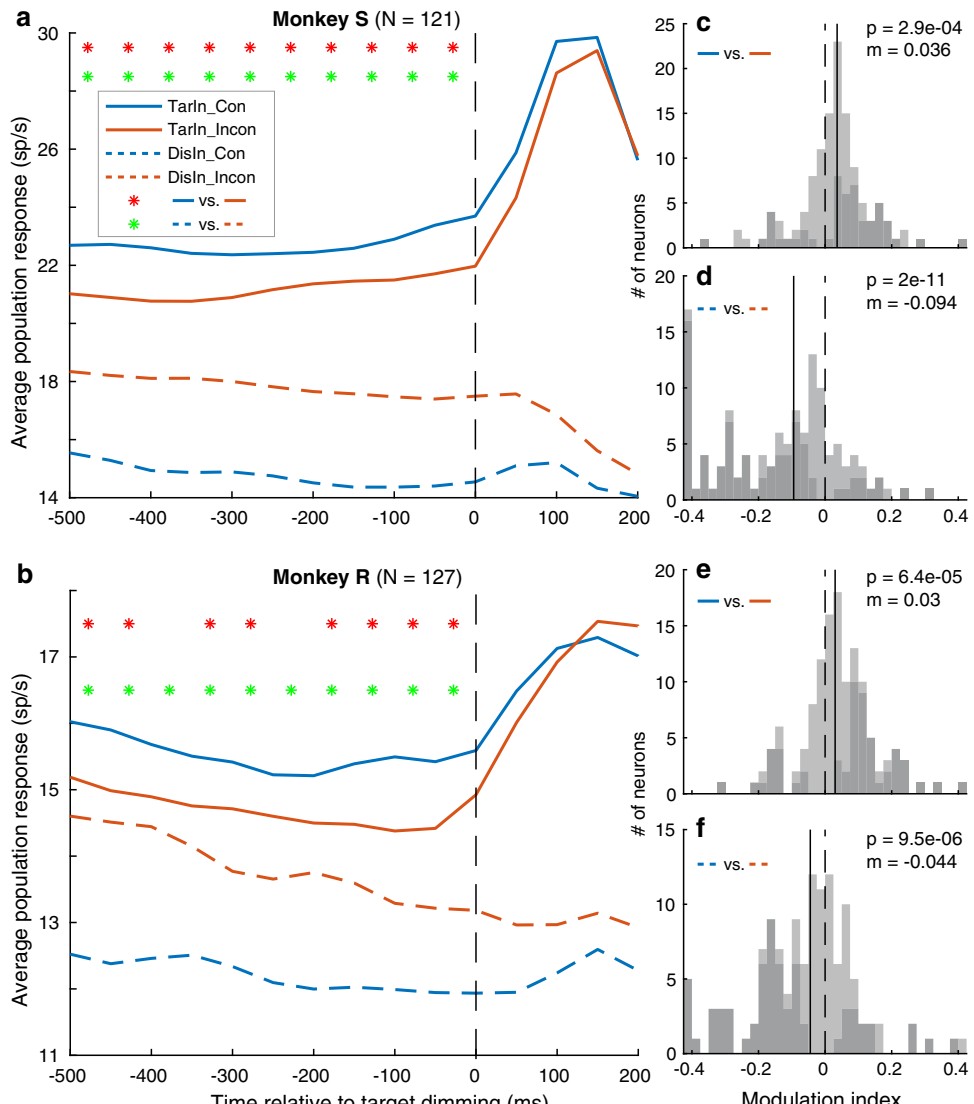

**Fig. 4 | Neuronal congruency effect in FEF. a, b** The population average peristimulus time histograms aligned to target dimming (vertical dashed line) indicate a signal decrease (difference between solid blue and orange lines) when targets fall inside the RF (TarIn), and a noise increase (difference between dashed blue and orange lines) when targets are presented outside the RF (DisIn) in incongruent compared to congruent conditions. Blue and orange lines represent congruent (Con) and incongruent (Incon) conditions, respectively. Red stars indicate significantly different responses (congruent versus incongruent) for successive, non-overlapping 50 ms bins for TarIn (i.e., significant signal change, two-tailed paired t-test, $p < 0.05$). Green stars indicate significantly different responses for DisIn (i.e., significant noise change). **c, e** Frequency histograms of modulation indices (MI = $(R_{con}-R_{incon})/(R_{con} + R_{incon})$), R: average response 500 ms prior target dimming) in monkey S **c** and monkey R **e** for TarIn. MI values > 0: signal increases more in congruent than incongruent trials. The p-value (Wilcoxon signed rank test, WSRT, two-tailed) and median MI values (m) are indicated. Black vertical dashed lines indicate no signal change; black lines the median MI. All values > 0.4 or < −0.4 are summed in the last and first data point of each histogram, respectively. Dark bars indicate neurons showing significantly different responses (two-tailed paired t-test, $p < 0.05$) between congruent and incongruent conditions. **d, f** Same as **c, e** but for DisIn. The distribution of MIs in **d** and **f** shows a predominance of values smaller than zero: noise decreases more in congruent than incongruent trials. Source data and the exact p values related to **a** and **b** are provided as a Source Data file.

analysis (Methods). All trials were categorized into four conditions based on conflict level (congruent or incongruent) and target locations relative to the RFs (TarIn or DisIn). Since we are mostly interested in the neuronal representations of the behavioral congruency effect, and not necessarily in conflict detection and resolution, we focused our analysis on a 500 ms time window immediately before the target dimming. This is the period closest to the behavioral response, when the conflict should have been detected and solved given the long delay. As expected and consistent with previous studies[47–49], FEF neurons were significantly modulated by covert spatial attention: average responses to congruent trials were higher for targets inside (solid blue lines) compared to outside (dashed blue lines) the neurons' RF (Fig. 4a, b). Attention increased the neuronal responses by a median value of 50%

in monkey S (Wilcoxon Signed Rank Test, WSRT, $p = 8.2e-13$) and 22.2% in monkey R (WSRT, $p = 4e-11$), which also reassured that the target stimuli were inside the neuron's RFs.

Critically, our data indicate that behavioral congruency effects can be explained by changing responses of both target (representing signal power) and distractor (representing noise power) neurons (Fig. 1a). Specifically, responses of target neurons (TarIn) were lower for incongruent (solid blue lines, Fig. 4a, b) compared to congruent conditions (solid orange lines), corresponding to a decrease in signal to target selection in incongruent trials. This signal decrease was consistently observed across subjects, and significant (two-tailed paired t-test, $p < 0.05$, red stars) at population level for all 50 ms bins 500 ms prior target dimming (except two bins in monkey R, Fig. 4a, b).

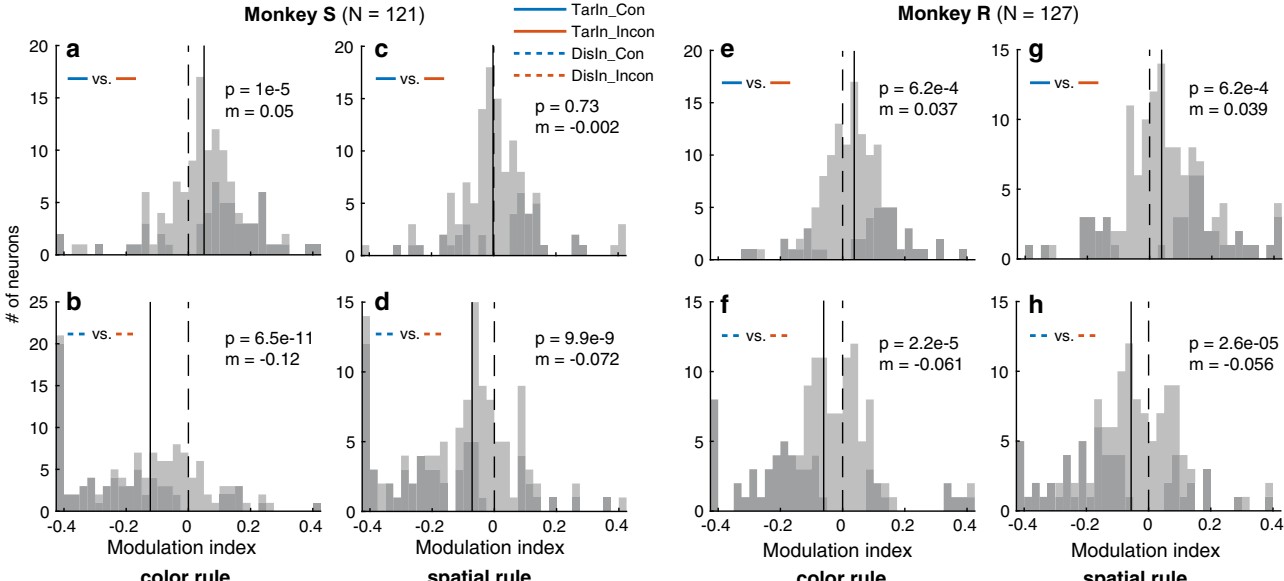

**Fig. 5 | Neuronal congruency effects under color- and spatial-rules.** An NCE is observed in trials of both rules in monkey S **a–d** and monkey R **e–h**. The conventions are the same as in Fig. 4c–f, but the color-rule **a**, **b**, **e**, **f** and the spatial-rule trials **c**, **d**, **g**, **h** are separated. **a**, **c**, **e**, **g** show the signal change between congruent and incongruent trials, and **b**, **d**, **f**, **h** show the noise change. All p values are from the two-tailed WSRT. Source data are provided as a Source Data file.

To quantify this effect at single neuron level, we calculated a modulation index ($MI = (R_{con} - R_{incon})/(R_{con} + R_{incon})$) for each neuron based on its average response to congruent ($R_{con}$) and incongruent ($R_{incon}$) conditions in the 500 ms interval preceding target dimming. The distribution of the MIs (Fig. 4c, e) significantly shifts to positive values (WSRT, monkey S: median = 0.036, $p = 2.9e\text{-}4$; monkey R: median = 0.03, $p = 6.4e\text{-}5$), indicating that more target neurons responded higher in congruent than incongruent conditions. 107 (43%) neurons showed significantly different responses between congruent and incongruent conditions (dark bars in Fig. 4c, e), from which 81 neurons (76%; i.e., 39 (77%) and 42 (75%) for monkey S and monkey R, respectively) responded significantly higher in congruent versus incongruent trials. Only 26 neurons (24%; i.e., 12 (24%) and 14 (25%) for S and R) showed the opposite effect.

On the other hand, the response of distractor neurons (DisIn) immediately before target dimming was higher, reflecting increased noise for incongruent (dashed orange lines) compared to congruent conditions (dashed blue lines). This increased noise in distractor neurons was consistent in both monkeys, and significant for every 50 ms bin 500 ms prior target dimming (green stars, two tailed paired t-test, p < 0.05). The distribution of MIs (Fig. 4d, f) shifted towards negative values (WSRT, S: median = −0.094, $p = 2e\text{-}11$; R: median = −0.044, $p = 9.5e\text{-}6$), indicating that more neurons responded higher in incongruent than congruent trials. 142 neurons (57%) showed significantly different responses between congruent and incongruent trials (dark bars in Fig. 4d, f), from which 116 neurons (82%; i.e., 66 (86%) and 50 (77%) for S and R) responded significantly higher during incongruent than congruent trials. Only 26 neurons (18%; i.e., 11 (14%) and 15 (23%) for S and R) showed the opposite effect.

In sum, our results indicate that the neuronal congruency effect is a combination of opposite response changes in target and distractor neurons, resulting in an SNR decrease to target selection within FEF's saliency map in incongruent compared to congruent conditions, confirming the third option in Fig. 1a.

### Consistent NCE for both spatial and color rule trials

The behavioral performance of the subjects was lower in incongruent compared to congruent conditions independent of the spatial or color rule, indicating that both the location and symbolic meaning of stimuli interfered with performance in incongruent trials. Hence an important question is whether both spatial and symbolic features modulate the neuronal responses in FEF in a congruency-dependent manner. In color-rule trials, subjects had to identify the color and translate it into a spatial location to shift their attention accordingly (i.e., they had to understand the symbolic meaning of the color). This is a more complex task and potentially involves different pathways compared to the spatial processing required during the spatial-rule trials. The latter trials might rely on the dorsal visual pathway, while color-rule trials might depend more on the ventral stream[50]. Therefore, the NCE under these two rules might be very different in FEF. We analyzed the signal and noise changes separately for both rules during a 500 ms interval before target dimming, exactly as in Fig. 4. Compared to congruent trials, both rules led to response decreases in target neurons (signal decreases) (Monkey S: Fig. 5a, c and Supplementary Fig. 2, monkey R: Fig. 5e, g) and response increases in distractor neurons (noise increases) during incongruent conditions (Monkey S: Fig. 5b, d, Monkey R: Fig. 5f, h). Hence, basic as well as complex features (e.g., the spatial location and symbolic meaning of cues in our task) can induce the behavioral and neuronal congruency effect.

Next, we examined whether the same neurons showed significant congruency effects under both rules. Surprisingly, this held true for the majority of neurons. Only few neurons (4% and 12% for TarIn, and 0% and 2% for DisIn trials, for monkey S and R, respectively) showed opposite results for the two rules (Fig. 6). Thus, our results suggest that the NCE within FEF does not depend on the rules used by the monkeys. Also note that approximately 25% of the neurons showed a higher response in incongruent compared to congruent TarIn trials (Fig. 6a, c), which may suggest the existence of different types of conflict processing neurons within FEF.

### NCE is dominated by congruency level

Next, we investigated whether the observed SNR differences can be explained by varying degrees of attentional deployment in congruent and incongruent conditions, since selective attention can also increase the SNR by enhancing the responses of target neurons and suppressing the responses of distractor neurons[51–53]. For example, it can be argued that subjects are more certain about the target location in congruent compared to incongruent conditions, which leads to more

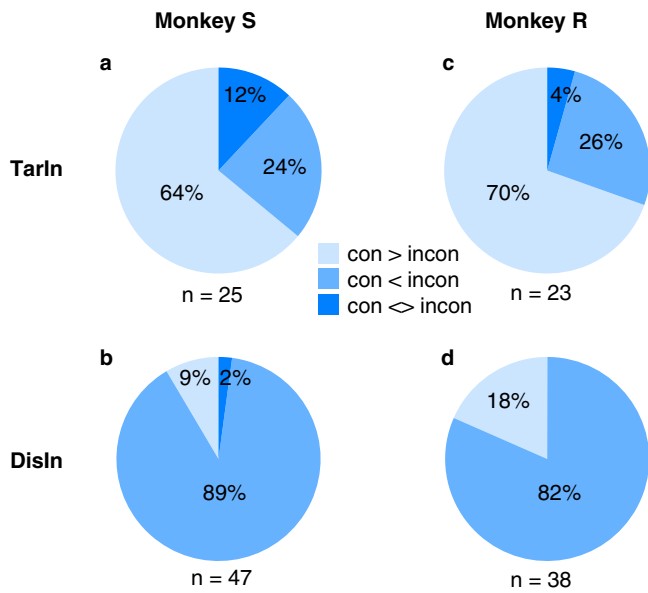

**Fig. 6 | Proportion of neurons showing congruency effects under both color- and spatial-rules.** When a target is in the RF **a**, **c**, the proportion of neurons showed significantly and consistently higher (70% and 64% for monkey S and R, light blue, con>incon) or lower (26% and 24% for monkey S and R, blue, con<incon) responses in congruent than incongruent conditions under both rules. Only 12% and 4% of neurons (monkey S and R) showed inconsistent congruency effects under both rules (dark blue, con<>incon). When the distractor is presented in the RF **b**, **d**, a proportion of neurons showed significantly and consistently lower (82% and 89% for monkey S and R, blue, con>incon) or higher (18% and 9% for monkey S and R, light blue, con<incon) responses in congruent than incongruent trials under both rules, only 0% and 2% of neurons (monkey S and R) showed inconsistent congruency effects in trials with different rules (dark blue, con<>incon). *N* indicates the number of neurons showing significant response difference between congruent and incongruent conditions under both rules. Source data are provided as a Source Data file.

focused attention on the target (Fig. 1b). In an attempt to investigate how attention affects SNR in the FEF between congruent and incongruent conditions in our experiment, we separated fast from slow trials for each condition based on the median RT of each condition for each recording session. Based on prior studies, we assume that subjects are more certain about the target location and/or attention is more focused on the target in fast compared to slow trials[42–46]. As expected, FEF neurons of both monkeys showed higher SNR in fast than slow trials in both congruent (Monkey S: Supplementary Fig. 3a, median SNR: 1.67 vs 1.41, $p = 5.7e-5$; Monkey R: Supplementary Fig. 3c, 1.23 vs 1.19, $p = 0.056$) and incongruent (Monkey S: Supplementary Fig. 3b, median SNR: 1.31 vs 1.08, $p = 2.2e-9$; Monkey R: Supplementary Fig. 3d, 1.10 vs 1.00, $p = 2e-5$) conditions, indicating that more focused attention indeed improves SNR within FEF.

More importantly, we compared RT and SNR of congruent and incongruent conditions in fast and slow trials, respectively. In both monkeys, we found higher SNR in the FEF in congruent than incongruent conditions for both fast (Monkey S: Fig. 7a, median SNR: 1.67 vs 1.31, $p = 7.2e-10$; Monkey R: Fig. 7d, 1.23 vs 1.10, $p = 7.8e-6$) and slow trials (Monkey S: Fig. 7b, 1.41 vs 1.08, $p = 3e-11$; Monkey R: Fig. 7e, 1.19 vs 1.00, $p = 2.2e-9$). However, we only found significantly faster RTs in congruent compared to incongruent conditions in slow trials (Monkey S: Fig. 7h, mean RT: 378 vs 388 ms, $p = 0.00049$; Monkey R: Fig. 7k, 390 vs 414 ms, $p = 0.00029$), but not in fast trials (Monkey S: Fig. 7g, mean RT: 320 vs 318 ms, $p = 0.30$; Monkey R: Fig. 7j, 340 vs 340 ms, $p = 0.55$). Secondly, we found that the RT in fast incongruent trials was much shorter than in slow congruent trials for both monkeys (Monkey S: Fig. 7i, mean RT: 318 vs 378 ms, $p = 0.00049$;

Monkey R: Fig. 7l, 340 vs 390 ms, $p = 0.00029$), suggesting that certainty was higher and/or attention was actually more focused on targets during fast incongruent than slow congruent trials. Surprisingly, however, SNR in fast incongruent conditions was significantly lower than in slow congruent trials for both monkeys (Monkey S: Fig. 7c, median SNR: 1.31 vs 1.41, $p = 0.00025$; Monkey R: Fig. 7f, 1.10 vs 1.19, $p = 8.4e-5$). These results suggest that the SNR in FEF is dominated by congruency rather than other signals that may affect RT.

The relatively large range of delays between color- and go cue that we used (660–1950 ms) provided us with the opportunity to investigate whether the NCE varies as function of delay duration. To this end, we separated short from long delay trials (target dimming occurred respectively 660–1260 ms, or 1261–1950 ms after color-cue offset). Note that no target dimming occurred until 660 ms after color-cue offset (Fig. 2).

In both monkeys, we found higher SNR in congruent than incongruent conditions for both short (Monkey S: Fig. 8a, median SNR: 1.57 vs 1.19, $p = 5.9e-11$; Monkey R: Fig. 8k, 1.17 vs 1.09, $p = 1.5e-4$) and long delay trials (Monkey S: Fig. 8b, median SNR: 1.50 vs 1.18, $p = 2.4e-9$; Monkey R: Fig. 8l, 1.28 vs 1.01, $p = 2.9e-11$). Since the amplitude of the neuronal response decreases as a function of time after color-cue offset (Supplementary Fig. 5), the average neuronal response before target dimming is significantly higher in short than long delay trials. This holds for all conditions and both monkeys, regardless of congruency level and irrespective whether a target or distractor is presented within the RF (all $p < 0.0005$, Fig. 8f–i, p–s). Despite this gradual decrease in activity, we did not observe consistent SNR differences between short and long delay trials for both congruent (Fig. 8c, 1.57 vs 1.50, $p = 0.2$) and incongruent (Fig. 8d, 1.19 vs 1.18, $p = 0.084$) conditions in Monkey S. In Monkey R, long delay trials show significantly higher SNRs than short delay trials for congruent conditions (Fig. 8m, 1.28 vs 1.17, $p = 0.041$), yet the opposite effect for incongruent conditions (Fig. 8n, 1.09 vs 1.01, $p = 0.0016$). These results suggest that the SNR in FEF is dominated by the congruency in the current study, and not by the duration of the delay (Fig. 8e, o), nor the amplitude of the neuronal response (Fig. 8j, t).

## Discussion

Using a task-switching paradigm, we found a behavioral congruency effect that can be linked to a neuronal congruency effect in FEF. The neuronal congruency effect comprises a signal change in target neurons and a noise change in distractor neurons, and their combination determines an SNR change within FEF's saliency map. The neuronal congruency effects are observed when either a simple (spatial) or complex stimulus feature (symbolic meaning) cues the animal.

Previous single unit studies in monkey frontal and parietal areas showed that, depending on the specific experimental conflict paradigm, individual neurons can respond either higher or lower during incongruent versus congruent trials. Therefore, these neurons were proposed to be involved in either detecting or resolving conflicting information[5,11,27,54–56]. However, these studies did not consider the neurons' visual RF location. We showed that the RF positions relative to target locations are crucial to explain neuronal congruency effects, at least when spatial features are important and/or the recorded neurons have a visual RF as in our paradigm. Neuronal response differences between congruent and incongruent conditions in previous studies[5,11,27,54–56] can be explained by the relative amplitude between signal (carried by target neurons) and noise (carried by distractor neurons). Specifically, if the amplitude of signal changes is higher than those of noise changes, the neurons will respond higher in congruent than incongruent conditions, and vice versa (See Fig. 1). Likewise, the neuronal congruency effects observed in visual FEF neurons can also generalize to motor neurons with specific motor fields. For example, saccade neurons in FEF and superior colliculus respond higher for pro-saccades (which can be considered as congruent trials) than anti-

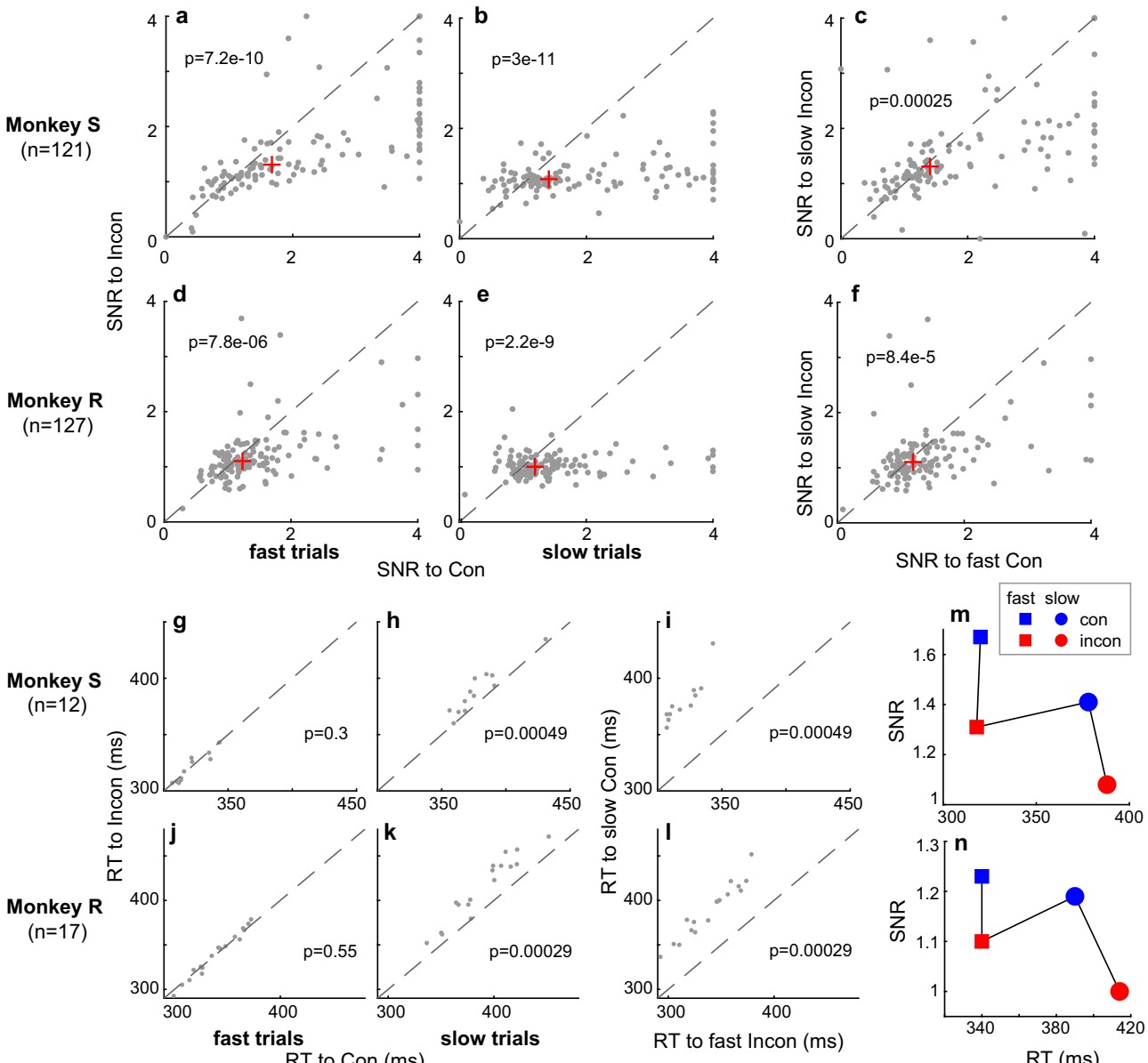

**Fig. 7 | The NCE is not determined by the RT.** More neurons show higher SNRs for congruent than incongruent conditions, both in fast **a**, **d** and slow **b**, **e** trials. RTs do not differ between congruent and incongruent fast trials **g**, **j**, and RTs are longer for incongruent slow trials **h**, **k**. Compared to fast incongruent trials, slow congruent trials yield longer RTs **i**, **l** but higher SNRs **c**, **f**. The SNRs are dominated by congruency rather than RT **m**, **n**. SNR values higher than 4 are plotted as 4. X and Y labels indicate the conditions being compared. Each dot in **a**–**f** and **g**–**l** represents a neuron and a recording session respectively. $SNR = R_{TarIn}/R_{DisIn}$, R: average response 500 ms prior target dimming. The dashed line represents the line of equality in each plot. The red plus (+) in **a**–**f** indicates the median SNR of the two conditions which are compared. All p-values are based on two-tailed WSRT (for SNR or RT comparisons). Source data are provided as a Source Data file.

saccades (mimicking incongruent trials), when the saccade is directed to the motor field of a neuron[11]. This result can be interpreted as a signal increase in congruent conditions, exactly as seen in our task. However, only neurons that were excited by the cues passed our selection criteria (Methods). Therefore, we do not know how neurons with complex RFs or that may be suppressed by the visual stimuli behave in our task. Future studies are required to address this interesting issue.

Human fMRI and EEG studies described higher responses in frontal cortex during higher conflict (incongruent) compared to lower conflict (congruent or neutral) conditions[5,13,14,57]. Due to the low spatial resolution of these techniques, it is impossible to distinguish target from distractor neurons in case they are homogeneously distributed within an area. Therefore, such results reflect an average population effect caused by congruency. For example, fMRI studies showed that

activity in brain areas processing task-relevant features was amplified in high conflict/incongruent compared to low conflict/congruent conditions[1,58]. The higher responses in incongruent conditions can be explained if the amplitude of the changes in noise is larger than the signal changes at populational level, and/or if more neurons show a noise change than a signal changes. Both predictions are confirmed by our and previous studies: FEF neurons showed stronger noise than signal changes, indicated by the median absolute modulation indices (Supplementary Fig. 4). Moreover, more neurons showed a significant noise than signal change (47% (116/248) versus 33% (81/248), Chi-squared test, $\chi^2 = 10.3$, $p = 0.001$). Also, another study revealed that more neurons showed a higher response during incongruent than congruent trials in the SEF[55]. Thus, single-unit results predict an overall increased response during incongruent compared to congruent trials in fMRI and EEG studies.

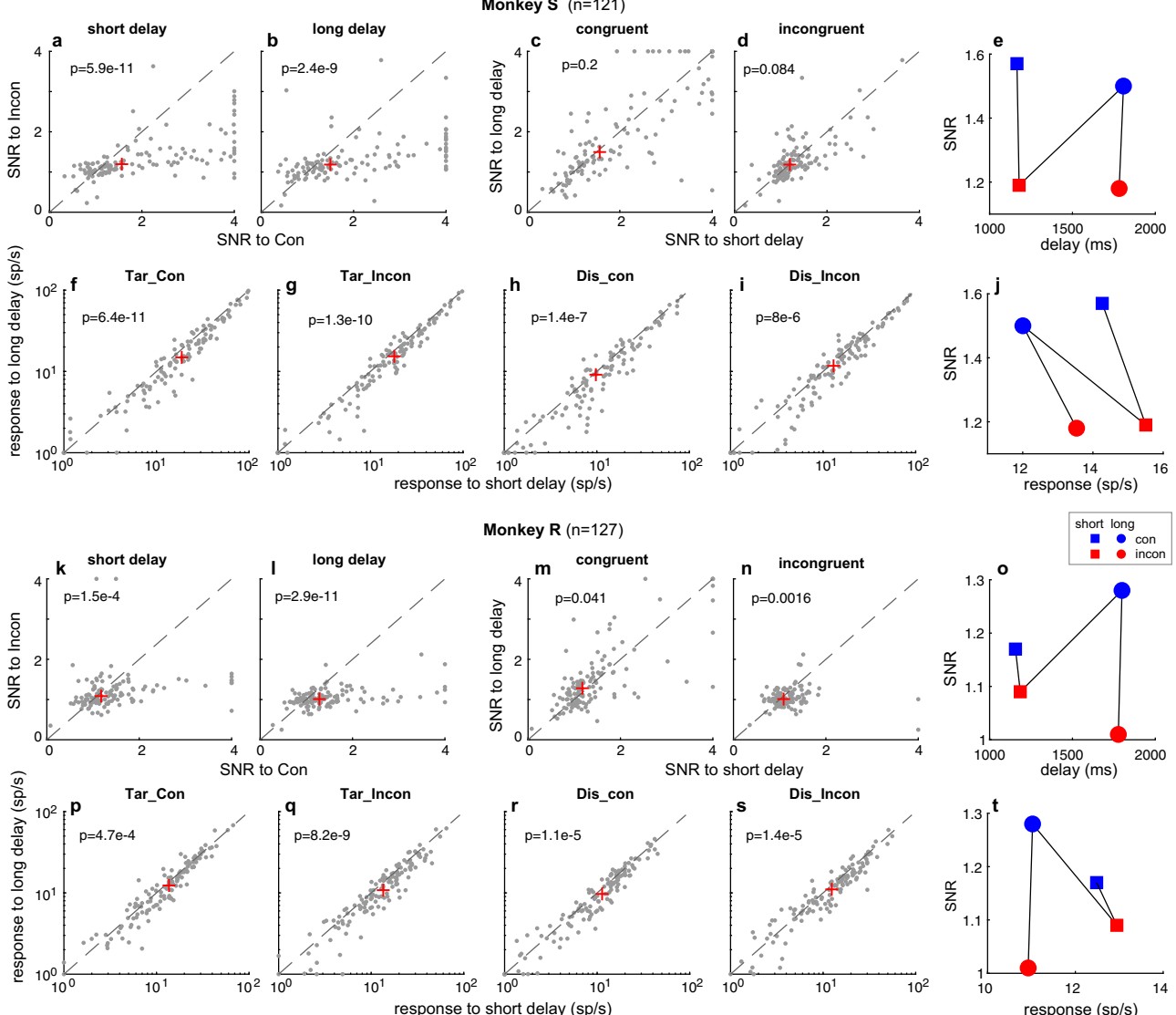

**Fig. 8 | The NCE is not determined by the delay between color cue and target dimming, nor by neuronal responsiveness.** More neurons show higher SNRs in congruent than incongruent conditions for both short **a**, **k** and long **b**, **l** delay trials. The SNR differences between short and long delay trials are not consistent across monkeys and congruency levels (**c**, **d**, **m**, **n**). The neurons in all conditions show higher responses in short compared to long delay trials (-500-0ms) before target dimming **f**–**i**, **p**–**s**. The SNRs are dominated by congruency rather than delay **e**, **o** or response amplitude **j**, **t**. The SNRs and p values are calculated as in Fig. 7. The other legends are the same as in Fig. 7. All p-values are based on two-tailed WSRT. Source data are provided as a Source Data file.

In the current study, we analyzed the neuronal responses immediately before target dimming (i.e. 500 ms prior the go-cue (target dimming) for the behavioral response), long after the conflict-inducing stimuli were shown. Visual input, motor output, RF positions, and attention allocation were matched between congruent and incongruent conditions during that time interval, leaving congruency as the major variable. In the trials we analyzed, the monkeys most likely focused their attention to the target well before target dimming because: i) the target location was determined by the rule- and color-cue in each trial, ii) there was a long delay between the color-cue and target dimming, which provided ample time to locate the target location, and iii) only correct trials were used in the analysis. Nevertheless, one may argue that other cognitive processes besides conflict processing differ between congruent and incongruent conditions. For example, the subject might be more certain about target location, or attention can be more focused on the target in congruent compared to incongruent trials. Consequently, the SNR of target selection within FEF would be higher in congruent conditions[42,43,59]. To investigate this possibility, we divided trials into slow and fast trials based on the median RT for each condition shown in Fig. 3. Then we calculated the SNR to congruent and incongruent conditions for fast and slow trials. We can reasonably assume that the subjects were more certain about the target location and/or focused more attention on the target in fast than slow trials[42,43]. If the SNR would be dominated by certainty/attention, or other cognitive processes that affect RT, we would have expected higher SNR in fast compared to slow trials (Fig. 1b top). On the other hand, if SNR is dominated by congruency, we would have expected higher SNR in congruent than incongruent trials (Fig. 1b bottom). Our data matched the latter hypothesis.

Moreover, the large range of delays between the color-cue and target dimming (go cue) enabled us to investigate the effect of delay duration on the NCE. We found a significant NCE for both long and short delay trials, which is consistent with a human behavioral study using a delayed match-to-sample Stroop task[60]. Surprisingly, however, we did not observe consistent differences in NCE between short and long delay trials (Fig. 8). Moreover, the neuronal response is

significantly higher in short versus long delay trials (Fig. 8). These results suggest that the NCE as indexed by the SNR before target dimming is dominated by the level of congruency, and that it is not determined by the RT (Fig. 7m, n), the length of the delay (Fig. 8e, o), and the responsiveness of the neurons (Fig. 8j, t). Yet, it may be sensitive to other cognitive processes that affect RT (Supplementary Fig. 3).

The NCE observed in the FEF support connectionist models such as the parallel distributed processing (PDP) model[12,61,62], which suggest that the interference from task-irrelevant information occurred automatically and was induced by prior learning and training[29,62]. Such mechanisms may be generalized to other conflict paradigms (also without spatial components as in the present task). For example, trials with the word RED printed in red during the Stroop task are congruent trials, while BLUE trials printed in red are incongruent trials. Reporting red would be correct in both trial types. Given our results, we predict that 'red-coding' (target) neurons will respond higher, while 'blue-coding' (distractor) neurons will respond lower in congruent trials, and vice versa in the incongruent trials. The same rationale can be generalized to most if not all conflict-related paradigms. Where the NCE emerges in the brain, what the relationship is with selective attention, and other related behaviors (such as microsaccades)[63], however, remains to be investigated in future studies.

## Methods

### Subjects and setup

Two adult male rhesus monkeys (*Macaca mulatta*, 6–8.5 kg, 8 and 10 years old during the period of recordings, respectively) participated in the current study. All experimental procedures and animal care were performed in accordance with the National Institute of Health's Guide for the care and use of laboratory animals, European legislation (Directive 2010/63/EU) and were approved by the Ethical Committee of KU Leuven. The animals were socially group-housed in cages between 16 and 32 m³ equipped with enrichment devices (toys, woods, ropes, foraging devices, etc.) at the primate facility of the KU Leuven Medical School. The animals were exposed to natural light and additional artificial light for 12 h every day. During the study, the animals had unrestricted access to food and daily access to restricted volumes of fruits and water. On training and experimental days, the animals were allowed unlimited access to the fluid through their performance during the experiments. Using operant conditioning techniques with positive reinforcers, the animals received fluid rewards for every correctly performed trial. Throughout the study, the animals' psychological and veterinary welfare was monitored daily by the veterinarians, the animal facility staff and the lab's scientists, all specialized in working with non-human primates. The two animals were healthy at the conclusion of our study and were subsequently employed in other studies.

Each monkey was implanted with an MRI-compatible head holder to minimize head movements during the training and recording. One standard recording chamber was also implanted in each monkey above the right frontal cortex to allow access to FEF, with implantation locations were chosen based on preceding MRI scans. The details of the implant surgery were previously described in Vanduffel et al.[64].

The experiments were performed in a dimly lit room with the only source of light being the display monitor. A Dell 17 inches LCD monitor at a distance of 57 cm from the monkeys' eyes was used to display the visual stimulus at a refresh rate of 60 Hz and a spatial resolution of around 40 pixels per degree. The monkeys were seated in a sphinx position in a custom-made primate chair, typically used for fMRI experiments[64]. Stimulus presentation, reward delivery, electrophysiological and behavioral data collection was controlled by custom software controlled by custom-built hardware and Dell Windows computers. The exact timing of the stimulus onsets and offsets was monitored by a photocell attached to the bottom-right corner of the LCD monitor. Eye-positions were monitored by an Iscan (Iscan, MA, USA) Infrared corneal reflection system at 120 Hz.

Neuronal activity was recorded extracellularly with Plexon 16-chanel V-probes (Plexon Inc., TX, USA). The 16 recording sites were aligned in a row with 150 μm inter-site spacing. The neuronal signal was filtered (300–1000 Hz), amplified, digitized, and stored with a TDT system (TDT Inc., TX, USA) with a 23 kHz sampling rate. All neuronal signals were recorded and stored for offline analyses. Offline spike sorting was performed with Plexon's Offline Sorter to isolate single and multi units. The FEF was identified by referencing the recordings to the structural MRI, in addition to the functional properties of the recorded neurons. Structurally, the recording sites, in the anterior bank of the arcuate sulcus, were localized with T1-weighted MRI imaging (TR = 2.5 s, TE = 4.35 ms, TI = 850 ms) (Supplementary Fig. 1). Functionally, the saccade direction and spatial tuning of the neurons was visually inspected online. A site was considered to be within the FEF, only when the neurons from at least 1 channel showed clear direction tuning for saccades and spatial tuning for visual stimuli (in all our recording sessions included in current study, we actually observed that the neurons from multiple channels showed tuning to spatial location and saccade directions). By combining the structural and functional evidence, we are confident that the locations we recorded from were in FEF.

### Behavioral tasks and stimuli

Once the recording 16-channel-probe (Plexon V-probe) arrived at target depth and the neuronal signals were stabilized, the monkeys first performed a fixation task whereby they maintained fixation on a central black fixation point (0.2 by 0.2 degrees). We identified the RF of neurons recorded from several channels by briefly flashing (200 ms) a white square stimulus on the gray background at one of 25 locations (5 by 5 grid, covering 25 by 20 degrees of the visual field). The RF location was determined by online inspection and analysis of the neuronal responses to the flashed stimuli. We could not map the RFs for the neurons recorded at all channels, since we focused on just a few channels during the recording, and the amount of trials that the subjects could perform was limited per day (preventing us to carefully map all the RFs from all neurons on all channels). After mapping the RFs from several channels, we selected that location covering most of the RFs based on the mapping results, for placing the target and distractor during the main task (Fig. 2). Next, we measured the neuron's saccade direction tuning by asking the monkeys to perform a visually guided saccade task from the center fixation point to a peripheral saccade target (7 visual degrees from the center fixation point). The saccade target was randomly picked from a set of 8 possible locations (evenly separated by 45° around a circle). We visually inspected the saccade direction tuning online. After identification of the location to position the stimuli, and part of the neurons showed tuning for saccade directions, we switched to the main task and recorded neuronal activity without further moving the V-probe (Fig. 2).

For the main experiment, the monkeys were trained to perform a task-switching paradigm (Fig. 2), where they were trained to pay covert attention to a target stimulus and respond to its dimming while ignoring a distractor stimulus. The monkeys initiated a trial by foveating a black fixation point (0.2 by 0.2 degrees) at the center of the screen. After 500 ms of fixation, the fixation point changed to a horizontal (0.4 by 0.2 degrees) or vertical (0.2 by 0.4 degrees) fixation bar which served as task rule-cue (color rule, or spatial rule, respectively) for the current trial. Accordingly, for a horizontal bar (color rule), the target stimulus was indicated by the color of the subsequently shown color-cue (which appeared 1100 ms after the task rule-cue, e.g., red or cyan for left, and pink or blue for right; please note that we used two pairs of color-cues (red-pink and cyan-blue) for monkey R (each

recording session only used one pair), and only one pair (red-pink) was used for Monkey S). The spatial location of the color-cue was irrelevant in these trials. Alternatively, when a vertical bar appeared (spatial rule), the target stimulus was indicated by the spatial location of the subsequent color-cue, its color being irrelevant. The rule-cue was presented for 500 ms, then, the original squared fixation point returned together with a pair of white peripheral stimuli (1 by 1 degree). The two stimuli were positioned at equal eccentricities, one of them was presented at the location determined by the RF mapping task within most of the recorded neurons' RFs, the other one at 180 degrees from the former (thus the two stimuli were central symmetrical, if one stimulus was presented in the top left quadrant, the second appeared in the bottom right quadrant). This alignment would maximize the distance between the two stimuli and ensure that one of them was within the recorded neurons' RF, while the other one was out (in present study, the two stimuli were separated by at least 14 visual degrees). After a delay of 600 ms following stimuli onset, one of the two white squares turned into a color-cue (1 by 1 degree) for 200 ms. Combined with the rule-cue, the target location was indicated either by the color (color-rule: horizontal bar; red and pink indicated that the target would be located at the left and right, respectively), or the location of the color-cue independent of its color (spatial rule: vertical bar).

The monkeys had to respond to a brief (150 ms) dimming in luminance of the target by pressing a button with their left hand (within 200–700 ms after the dimming). Target dimming occurred between 660 and 1950 ms after color-cue offset in every trial. Moreover, to ensure that the subjects were attending to the target rather than responding to any dimming, the subjects had to ignore similar dimming of the distractor, which happened randomly in 50% of the trials, and never more than once in a trial. Distractor dimming occurred between 200 and 1500 ms after color-cue offset, with the additional requirement that it happened at least 300 ms before target dimming. This separation ensured that the monkeys' responses to the distractor dimming could be identified and distinguished from their responses to the target dimming. Trials terminated 700 ms after the target dimming, and the monkey received a drop of juice if the button had been correctly pressed during this period. Note that there was a target dimming in each trial, so the monkey was required to make an identical operant response in each trial in order to be rewarded, thus we excluded a stimulus-response conflict. During the task, the background was always gray (RGB values: 70, 70, 70; 4 cd/ $m^2$); the fixation point and the task rule-cue (horizontal and vertical fixation point) were black (RGB: 0, 0, 0; 0.1 cd/$m^2$); the squared stimuli were white (RGB: 255, 255, 255; 77 cd/$m^2$); the dimming of the squared stimuli corresponded to a gray stimulus (RGB: 210, 210, 210; 51 cd/$m^2$). Monkeys had to maintain fixation within a (virtual) squared window of 2.5–3 visual degrees centered around the fixation point until they received the reward. Please note that in the paradigm, between the congruent and incongruent conditions, we controlled (1) the visual input of the stimuli, by analyzing the neuronal response to exactly the same stimuli in the delay period; (2) the response of the subjects by asking the subjects performing the same response in all trials; 3) the allocation of spatial attention by requiring subjects to attend the target stimulus within or out of the neurons' RF.

**Data analysis**

The behavioral and neuronal data analysis was performed using MATLAB (MathWorks, MA, USA). The MATLAB codes and data related to all the figures are provided in Supplementary Data 1. We performed 12 recording sessions in Monkey S, and 17 recording sessions in Monkey R, and all the sessions included 32–40 correct trials for each condition shown in Fig. 3. The correct trials (hits) corresponded to trials in which the button was pressed within 200-700 ms after target dimming. Incorrect trials included all false alarm

trials (i.e., when the monkeys pressed the button at the wrong time) and missed trials (the monkeys did not press a button within the 200–700 ms response window after target dimming). All trials during which fixation was interrupted were excluded from the analysis. The performance of each session was defined as the number of correct trials divided by the sum of the number of correct and incorrect trials.

Neuronal activity was recorded from the FEF in the right hemisphere using Plexon's 16-channel V-probes. The spikes were offline sorted into single- and multi-units using Plexon's offline sorter. A total of 591 single neurons (267 from Monkey S, 324 from Monkey R) were isolated using offline sorting. Since our design required that one of the two stimuli should be presented in the neurons' RF, and not all of the neurons satisfied with this since: (1) some of the neurons were not visually driven (by these white flashing squares) in FEF, (2) multiple neurons were recorded with the probe at the same time, some of their RFs did not cover the target nor the distractor. Therefore, we first identified the visually-driven neurons that were activated by the white squared stimuli before the color cues. A neuron was qualified as visually-driven when it showed a significantly higher response in the 0–500 ms time window after the onset of the two stimuli onset compared to the response in 200–500 ms after onset of the fixation point. (two-tailed paired t-test, $p < 0.05$) in all correct trials. Next, for these visually-driven neurons, to determine their RF at the left or right visual hemifield, we analyzed the neuronal response induced by the target dimming. Since we recorded from the right FEF, we would expect most of the neurons' RFs were in the left visual hemifield[65]. We only included those neurons in our further analysis that showed a significantly higher response to the target dimming in the left compared to the right hemifield (50–200 ms after target dimming onset, two-tailed paired t-test, $p < 0.05$). Therefore, the RFs of the selected neurons would cover targets presented in the left visual hemifield, while targets presented in the right visual field were outside the neuron's RFs, which was confirmed by the neuronal response in Fig. 4. We found a clear response to the target dimming when the target was supposed to be inside the RF (solid lines), while the response to the target dimming was not clear when the target was supposed to be out of the RF (dashed lines). Using the additional criteria, we were able to select 248 visual neurons with pronounced contralateral (left) RFs (121 from Monkey S, 127 from Monkey R) for further analysis.

Peri-stimulus time histograms (PSTHs) were calculated by smoothing the data with a Gaussian-weighted moving average over each 50 ms time bins in a window of 200 ms. The average activity across trials was first calculated for each neuron, and then, to obtain the displayed PSTHs in Fig. 4a, b, the PSTHs of individual neurons were averaged across all neurons. For the bin-by-bin statistical significance tests (Fig. 4a, b), we performed a paired t-test across neurons for each given bin based on the non-smoothed average response of each neuron, if a test between two conditions was significant/non-significant for a given bin, we assumed that it was significant/non-significant for the entire duration of the 50 ms bin. To avoid that the transient response to the brief dimming of the distractor stimulus affected the results, for the trials including a distractor dimming, we excluded the period of 0–200 ms following the distractor dimming onset from the PSTH and firing-rate calculations. For the differences between two conditions (Fig. 4, Supplementary Fig. 2) at neuron level, we reported the modulation effects using a modulation index (MI) for each neuron, which was defined as the difference in the firing-rates for the two conditions divided by their sum for a time window of 500 ms before target dimming onset. We used the median value of the MIs of all neurons to report the population modulation effect, and used a WSRT to test whether the effect was significant. The same analysis was performed for trials with a color-rule and a spatial-rule (Fig. 5).

For the SNR analysis of fast and slow trials, we first divided all the trials in each recording session into fast and slow trials based on the median RT for each condition shown in Fig. 4, i.e., the fast and slow trials for TarIn_con, TarIn_incon, DisIn_con, and DisIn_incon. The avarage neuronal response of the 500 ms before target dimming to TarIn_con ($R_{TarIn\_con}$) and TarIn_incon ($R_{TarIn\_incon}$) represented the 'signal' in congruent and incongruent trials, respectively, while the $R_{DisIn\_con}$ and $R_{DisIn\_incon}$ represented the 'noise'. Then, the SNR of the FEF neurons for the congruent and incongruent trials was calculated as $R_{TarIn\_con}/R_{DisIn\_con}$ and $R_{TarIn\_incon}/R_{DisIn\_incon}$ respectively for each recording session. For each neuron, we calculated the SNR for 4 conditions, i.e., the fast_congruent condition, fast_incongruent condition, slow_congruent condition, slow_incongruent condition. We compared the SNR differential based on these 4 conditions to investigate the relationship between NCE and RT (Fig. 7). The SNR analysis for short and long delay trials are similar. We first divided all the trials in each recording session into short and long trials based on the delay between color cue and target dimming for each condition. The short and long delay trials had a delay range of 660–1260 ms and 1260–1950 ms, respectively. Then, we calculated the SNR for 4 conditions, i.e., short_congruent condition, short_incongruent condition, long_congruent condition, long_ incongruent condition. We compared the SNR differential based on these 4 conditions to investigate the relationship between NCE and delay (Fig. 8).

### Reporting summary

Further information on research design is available in the Nature Research Reporting Summary linked to this article.

## Data availability

Source data are provided in this paper.

## Code availability

Code are provided with this paper.

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

## Acknowledgements

We thank C.F., A.C., P.K., I.P., C.U., A.H., G.M., M.D., W.D. and S.V. for technical and administrative support. We thank T.V. for his comments on the manuscript. This work received funding from KU Leuven C14/17/109, C14/21/111, IDN/20/016; Fonds Wetenschappelijk Onderzoek-Vlaanderen (FWO-Flanders) G0B8617N, G0E0520N, G0C1920N; the European Union's Horizon 2020 Framework Programme for Research and Innovation under Grant Agreement No 945539 (Human Brain Project SGA3) to W.V., and from Fonds Wetenschappelijk Onderzoek-Vlaanderen (FWO-Flanders) 1501320 N, 12W0919N to T.Y. T.Y is postdoctoral fellow of the FWO-Flanders.

## Author contributions

Conceptualization, T.Y., and W.V., Methodology, T.Y., and W.V., Investigation, T.Y., Writing, T.Y., and W.V., Funding acquisition, T.Y., and W.V., Supervision, W.V.

## Competing interests

The authors declare no competing interests.
