## [Peer Review File · Nature Communications]

Neuronal congruency effects in macaque prefrontal cortexREVIEWER COMMENTS

Reviewer #1 (Remarks to the Author):

In this manuscript entitled 'Neural congruency effects in macaque frontal cortex', Yao and Vanduffel record from the prefrontal cortex (frontal eye fields) of macaque monkeys while these are performing a task switching paradigm involving conflict between sensory information and task instruction. The goal is to precisely characterize the neural bases of behavioral congruency effects observed when task-relevant and task-irrelevant stimulus features induce conflicts that impair behavioral performance.

The major novelty of this work is that rather than focusing on conflict detection or resolution, the study identifies the implicit neuronal signatures induced by such sensory conflict. The work is very interesting and has great value beyond monkey electrophysiologists as it contributes to a better understanding of non-invasive neurophysiological signatures of conflict. I however have major concerns I would like to see addressed before this work can be considered for publication.

Major:

1. The introduction introduces the notion of SNR. This SNR is defined in the legend of Figure 1 and is a relatively idiosyncratic definition. This should be better explained in the introduction itself specifying what this SNR refers to as well as (and quite crucially) what it does not refer to amongst the multiple classical electrophysiological definitions of SNR (e.g. spiking neuronal variability, noise correlation, etc).
2. The authors state very clear how congruency is interpreted for the color-rule trials but take it for granted that congruency in the spatial trials is clear. I acknowledge that the (implicit) information about this can be gathered throughout the manuscript. I however strongly recommend that this is actually made out clear and explicit right at the beginning of the manuscript, thus unpacking the relationship between task-relevance and conflict in each of the color and spatial trials. (see also my minor comments in this respect).
3. The FEF neurons have very complex receptive fields. First some will be excitatory and others will be inhibitory. Based on the neuronal selection criteria of the authors (recordings were performed in the right hemifield, so authors only including neurons for which target dimming induced a stronger response on the left than on the right). As a result, they mostly selected excitatory neurons. This might be worth noting. Most importantly, the general SNR framework presented in figure 1 assumes that higher neuronal responses == better target representation, including in the task-irrelevant neurons. This assumes that all neuronal modulations are excitatory modulations relative to the individual neuron's baseline. However, due to the complexity of their receptive fields, FEF neurons often show inhibitory responses to visual stimuli AND/OR attention orienting (visual and attention spatial selectivities not always strictly matching, especially for the inhibitory component). As a result, some of the task-irrelevant neurons might not show higher responses in incongruent condition (middle panel, figure 1), but rather lower inhibition in incongruent condition (thus actually less signal). Such situations are expected based on the observation of the distribution of the modulation indexes in Figure 4. While these MI do not exactly report what I addressing here, these distributions do show that the cell sample

displays complex modulations to the rule+cue combination including situations where incongruent conditions lead to a higher firing rate than the incongruent condition. Overall, and as a result, all throughout the manuscript, the TargetOut plots might actually represent a sum of two different task-irrelevant populations: one the activity of which is enhanced relative to baseline during incongruent condition and one whose activity is actually suppressed (though negatively). What would solve this issue would be to quantify pre-target activity changes relative to pre-rule-cue baseline. While I do understand that as neurophysiologists we often report the central most trend, the authors here propose a model of prefrontal cortex conflict resolution. It is important that they can rule out alternative conditions as I describe here and that are not considered in figure 1, as well as rule out the possibility of a co-existence of computational models in the FEF.

4. The authors show that the described neuronal congruency effects are not fully accounted for by reaction times (RTs). This is quite interesting. I however have several comments here. First, and as appropriately done in the result section, the authors are very cautious in assigning fast and slow RT to attention orientation, as RT differences can arise from multiple underlying brain mechanisms. However, at multiple other locations, including in the abstract, results and discussion, this caution disappears. I would stick to a more pragmatic description of the RT effects without exclusively assigning them to attention.

5. Second, important analyses are based on trial classification in terms of fast and slow RT. Can the authors show the raw RT data for each experimental condition? There are two underlying questions behind this: 1- does the RT sample contain express RTs? 2-how variable are RTs across conditions and what is the effect of this variability on the reported neuronal analyses?

6. Third (echoing to 5.2), can you please provide RTs for the data presented in figure 3.

7. How are congruence effects matching for individual neurons across rules? For example, figure 4&5 shows congruency effects under the color and spatial rules at the population level, but doesn't address whether the congruency effect under the spatial rule predicts that under the color rule, or whether some cells are dominated by the spatial components, others by the task relevance and others yet by the color etc. Such a diversity would be expected in the general context of mixed selectivity coding in the prefrontal cortex.

8. As discussed in point 3, SNR analysis should be performed computing SNRs relative to neuronal baseline.

Minor:

1. The frontal eye field is strictly speaking at best in the prefrontal cortex although some might even argue that its cellular organization is more alike the premotor cortex than the prefrontal cortex. In any case, the term frontal cortex used in the paper is misleading and I would recommend changing it for prefrontal.
2. Figure 1. Remove (A) and (B) from legend header as these are inaccurate.
3. Figure 1. Small box with colored squares and incon/con labels is missing information on task relevance

4. Figure 1. Please make it clear both in the figure and the text 1) that congruence is defined in terms of match between instruction and RF spatial location and 2) only applies to the rule 1 condition (implicit in figure but should be made explicit)
5. Please indicate distractor dimming in figure
6. Page 6, line 19, please cite recent Di Bello et al. (2021, Cerebral Cortical) paper of task-relevant and task irrelevant distractor coding in the FEF.
7. Indicate in figure S2 where (what timing) on panel A indexes presented in panels B and C were computed.

Reviewer #2 (Remarks to the Author):

The study by Yao and Vanduffel on “Neuronal congruency effects in macaque frontal cortex” tackles a very important issue, that is what the neural basis of congruency effects in task switching paradigms such as the Stroop task is. Such tasks that induce interference by means of congruency/incongruency conditions have been widely used in psychology and the study of cognitive control. Yet, surprisingly little is known about their neural basis.

This study has a large number of significant strengths:

- 1) Probing two versions of the task: The authors designed a novel congruency task that they probed in two versions, a color and a spatial version, demonstrating converging results and thus generalization of their neural results.
- 2) Behavioral results: The congruency effects are very impressive with this task. My only minor concern is here that performance in the congruent condition is close to ceiling, suggesting that the dimming was quite salient. Was there a particular rationale for having the animals perform close to ceiling?
- 3) Neural results: The neural results are very clear supporting the authors’ hypothesis-driven rationale of target-related increases in signal (i.e. at the attended location) and decreases of noise (at the unattended location) as a function of congruency.
- 4) Link to behavior: The neural results are also able at least partially to account for behavior.

This is a very clearly written and presented study on an important topic that reports straightforward results that will add nicely to the literature.

I have only a few comments that the authors may address (in addition to my questions above):

- Temporal structure: The delay period is quite long. Did you mix trials with shorter delay (e.g. <1s) with those that have longer delays (e.g. >1s)? It would be interesting to look at these trials separately to understand when the spatial location is decided and how well spatial attention is sustained over time. Investigating the 500ms time window before target dimming is a good start, but there could be large differences between short and long trials that influence the main effects that are presented.

- Microsaccades: Is it possible that microsaccades known to modulate neural activity across the attention network contributed to the neural effects? MSs would be executed systematically towards or away from the RF in some conditions and could also influence incongruency effects.

- Rationale for FEF: Please provide a better rationale for FEF. Since FEF neurons do not care about features (just about their spatial implementation), it is perhaps not so surprising that there were no differences between the color and spatial conditions. It is unlikely that the color cue was implemented in FEF. Please discuss how the cueing paradigm may be implemented in FEF, particularly with regard to the results of feature-based attention in PFC reported by the Desimone lab in recent years.

We thank both reviewers for their positive criticism and insightful suggestions, which helped us improve the revised manuscript. We introduced the acronym 'NCE' to avoid the repetitive use of 'neuronal congruency effect'. Inspired by the comments of the reviewers, we changed 'task-relevant and task-irrelevant neurons' to 'target and distractor neurons', which are the neurons processing the target and distractor, respectively. This terminology should reduce confusion with task-relevant and task-irrelevant information. Accordingly, we replaced 'TarOut' by 'DisIn'. We also changed the layout of Figure 5, and updated Figure 7 (original Figure 6). Some of the SNR and p values are slightly different from the previous version in the results section (related to Figure 7) due to a tiny bug in the analysis program that we discovered while re-analyzing some of the results. However, this does not change any conclusion in any way.

Reviewer #1 (Remarks to the Author):

In this manuscript entitled 'Neural congruency effects in macaque frontal cortex', Yao and Vanduffel record from the prefrontal cortex (frontal eye fields) of macaque monkeys while these are performing a task switching paradigm involving conflict between sensory information and task instruction. The goal is to precisely characterize the neural bases of behavioral congruency effects observed when task-relevant and task-irrelevant stimulus features induce conflicts that impair behavioral performance.

The major novelty of this work is that rather than focusing on conflict detection or resolution, the study identifies the implicit neuronal signatures induced by such sensory conflict. The work is very interesting and has great value beyond monkey electrophysiologists as it contributes to a better understanding of non-invasive neurophysiological signatures of conflict. I however have major concerns I would like to see addressed before this work can be considered for publication.

Major:

1. The introduction introduces the notion of SNR. This SNR is defined in the legend of Figure 1 and is a relatively idiosyncratic definition. This should be better explained in the introduction itself specifying what this SNR refers to as well as (and quite crucially) what it does not refer to amongst the multiple classical electrophysiological definitions of SNR (e.g. spiking neuronal variability, noise correlation, etc).

We thank the reviewer for the suggestion. We explained SNR more explicitly in the introduction (line72-81): "The response of target and distractor neurons, corresponding to 'signal' and 'noise' in the context of target representations, have a positive and negative effect on target representations, respectively (Fig. 1a). Therefore, the signal-to-noise ratio (SNR, signal divided by noise) in the current study reflects how well the target can be distinguished from a distractor by a neuron or an area (e.g., FEF). In other words, it reflects how well the target is represented. Please note that the SNR is not related to net responses, spiking variability or noise correlations. We predict that the SNR of a target representation within FEF (carried by both target and distractor neurons) is lower in incongruent than congruent conditions, which can result in a behavioral congruency effect (Fig. 1a, top). Thus,

the larger the difference in SNR between congruent and incongruent trials (Δ SNR in Fig. 1a), the larger the behavioral congruency effect.”

2. The authors state very clear how congruency is interpreted for the color-rule trials but take it for granted that congruency in the spatial trials is clear. I acknowledge that the (implicit) information about this can be gathered throughout the manuscript. I however strongly recommend that this is actually made out clear and explicit right at the beginning of the manuscript, thus unpacking the relationship between task-relevance and conflict in each of the color and spatial trials. (see also my minor comments in this respect).

We apologize for the confusion and we added several sentences in the first paragraph of the Results section (line 133-143) to explain how conflict can arise under both rules: “A trial was considered congruent when the color and spatial location of the color-cue indicated the same target location under both rules (Supplementary Table 1). Otherwise, the trial is incongruent. A conflict arises during incongruent trials since two stimulus features of the color-cue (i.e., its color and spatial location) indicate different target locations. Yet, only one of them is task-relevant as indicated by the task-rule cue. For example, when a pink color-cue is presented on the left side during color-rule trials, the target will be on the right side. In this case, we can investigate how task-irrelevant cue location interferes with behavioral performance, similarly as in the Simon task. During spatial-rule trials, however, a pink cue on the left would be indicative for a target on the left. In that case, we can investigate how the symbolic meaning of the cue (pink = right in the color rule trials) interferes with behavioral performance, similarly as in a Stroop task.”

3. The FEF neurons have very complex receptive fields. First some will be excitatory and others will be inhibitory. Based on the neuronal selection criteria of the authors (recordings were performed in the right hemifield, so authors only including neurons for which target dimming induced a stronger response on the left than on the right). As a result, they mostly selected excitatory neurons. This might be worth noting. Most importantly, the general SNR framework presented in figure 1 assumes that higher neuronal responses == better target representation, including in the task-irrelevant neurons. This assumes that all neuronal modulations are excitatory modulations relative to the individual neuron’s baseline. However, due to the complexity of their receptive fields, FEF neurons often show inhibitory responses to visual stimuli AND/OR attention orienting (visual and attention spatial selectivities not always strictly matching, especially for the inhibitory component). As a result, some of the task-irrelevant neurons might not show higher responses in incongruent condition (middle panel, figure 1), but rather lower inhibition in incongruent condition (thus actually less signal). Such situations are expected based on the observation of the distribution of the modulation indexes in Figure 4. While these MI do not exactly report what I addressing here, these distributions do show that the cell sample displays complex modulations to the rule+cue combination including situations where incongruent conditions lead to a higher firing rate than the congruent condition. Overall, and as a result, all throughout the manuscript, the TargetOut plots might actually represent a sum of two different task-irrelevant populations: one the activity of which is enhanced relative to baseline during incongruent condition and one whose activity is actually suppressed (though negatively). What would solve this

issue would be to quantify pre-target activity changes relative to pre-rule-cue baseline. While I do understand that as neurophysiologists we often report the central most trend, the authors here propose a model of prefrontal cortex conflict resolution. It is important that they can rule out alternative conditions as I describe here and that are not considered in figure 1, as well as rule out the possibility of a co-existence of computational models in the FEF.

The reviewer is correct that the receptive fields of the FEF can be very complex. Hence, this comment is quite relevant. However, all the neurons that passed our selection criteria were excited by the visual stimuli. This was confirmed by a new analysis in which we compared the neuronal response in the 400 ms interval prior to the task-rule cue (baseline) and the 500 ms interval immediately after stimulus onset (visual response) for each condition. The results indicate that the majority of neurons increased their response after stimulus onset (Figure RR1, end of this document). We added some sentences in (line 373-376) to highlight this additional analysis: “However, only neurons that were excited by the cues passed our selection criteria (Methods). Therefore, we do not know how neurons with complex RFs or neurons that may be suppressed by the visual stimuli behave in our task. Future studies are required to address this interesting issue.”

As already suggested in point 1, it appeared that SNR as we used it in the present manuscript was not clearly explained. SNR reflects how well the target can be distinguished from a distractor by a neuron or an area (e.g. FEF): see detailed reply to reviewer’s point 1 and adapted figure 1. The SNR is determined by the relative response of neurons to targets and distractors. In fact, the baseline response of the neurons is irrelevant. A higher response of the target neurons (the neurons with a target presented in their RF) or distractor neurons alone does not necessarily result in a higher SNR or better target representation. The most important information we aim to deliver with Figure 1 is how congruency may affect the neuronal responses: It can affect the target neurons and not the distractor neurons (signal change, left panel). Alternatively, it can affect the distractor neurons but not the target neurons (noise change, middle panel), or both (right panel). We now explicitly explain the meaning of signal change (Δ signal), noise change (Δ noise) and SNR change (Δ SNR) between congruent and incongruent conditions in Figure 1.

Interestingly, the neuronal response in short delay trials is higher than in long delay trials (see new analysis). This is because the neuronal response systematically decreased as a function of time after cue offset (supplementary Figure 5, Figure 8). The neuronal congruency effect (NCE, indicated by SNR), however, is significant in both short and long delay trials (Figure 8). This result suggests that the NCE is not determined by the neuronal response amplitude.

It needs to be noted, however, that ~25% of the target neurons showed an “opposite” congruency effect (and 10-20% of the distractor neurons) (detailed in point 7 and Figure 6). We discussed this in line 262-264 “Also note that approximately 25% of the neurons showed a higher response in incongruent compared to congruent Tar trials (Fig. 6a, b), which may suggest the existence of different types of conflict processing neurons within FEF.”

4. The authors show that the described neuronal congruency effects are not fully accounted for by reaction times (RTs). This is quite interesting. I however have several comments here. First, and as appropriately done in the result section, the authors are very cautious in assigning fast and slow RT to attention orientation, as RT differences can arise from multiple underlying brain mechanisms. However, at multiple other locations, including in the abstract, results and discussion, this caution disappears. I would stick to a more pragmatic description of the RT effects without exclusively assigning them to attention.

We fully agree with the reviewer that RT differences can be caused by different mechanisms. We aligned our statements regarding this point throughout the manuscript (including the abstract, the introduction, the results, and the discussion section).

In Abstract (line 23-25): “Such NCE is dominated by the level of congruency, and is not determined by the task rules the subjects used, their reaction times (RT), the length of the delay period, nor the response levels of the neurons.”

In introduction (line 118-121): “Reaction times (RTs) are sensitive to several cognitive operations. For example, they are faster when subjects are more certain about the location of a target, or when attention is more focused on the target⁴²⁻⁴⁶. To test whether the NCE is also affected by other cognitive modulations that affect RT, we compared the NCE in trials with fast and slow RTs.”

In discussion (line 401-404): “one may argue that other cognitive processes besides conflict processing differ between congruent and incongruent conditions. For example, the subject might be more certain about target location, or attention can be more focused on the target in congruent compared to incongruent trials”.

and line 409: “If the SNR would be dominated by certainty/attention, or other cognitive processes that affect RT,…”.

and line 418-422: “These results suggest that the NCE as indexed by the SNR before target dimming is dominated by the level of congruency, and that it is not determined by the RT (Fig. 7m, n), the length of the delay (Fig. 8e, o), and the responsiveness of the neurons (Fig 8j, t). Yet, it may be sensitive to other cognitive processes that affect RT (Supplementary Fig. 3).”

5. Second, important analyses are based on trial classification in terms of fast and slow RT. Can the authors show the raw RT data for each experimental condition? There are two underlying questions behind this: 1- does the RT sample contain express RTs? 2-how variable are RTs across conditions and what is the effect of this variability on the reported neuronal analyses?

The distributions of raw RTs for each condition argue against express RTs (Figure RR2). Secondly, as shown in Figure 7, RT does not determine the NCE. Yet, it is plausible that SNR in the FEF can be modulated by cognitive processes that also affect RT (Supplementary Figure 3).

6. Third (echoing to 5.2), can you please provide RTs for the data presented in figure 3.

The figures and related legends for the RTs are added to Figure 3. We also provided a supplementary table 3 for the statistical results of each comparison. In the results

section, we added the following (line 148-153): “We also found a small but significant average RT difference between congruent and incongruent conditions (Fig. 3c, d, the last pair of bars, average: monkey S: 349 vs. 353ms; $t_{11}=2.85$, $p=0.016$; monkey R: 365 vs. 377ms, $t_{16}=7.1$, $p=2.5e-6$). It is important to note that, unlike traditional conflict tasks, we did not employ a reaction time task. In our study, there was a variable and relatively long gap between the conflict-inducing cue (the color cue) and the go cue (target dimming). Hence monkeys were not forced to make a fast response.”

7. How are congruence effects matching for individual neurons across rules? For example, figure 4&5 shows congruency effects under the color and spatial rules at the population level, but doesn't address whether the congruency effect under the spatial rule predicts that under the color rule, or whether some cells are dominated by the spatial components, others by the task relevance and others yet by the color etc. Such a diversity would be expected in the general context of mixed selectivity coding in the prefrontal cortex.

We thank the reviewer for raising this very interesting point. We performed a new analysis to examine whether the same neurons show a congruency effect under both rules. We found the answer is yes. We detailed the results in line 258-264 and Figure 6. “Next, we examined whether the same neurons showed significant congruency effects under both rules. Surprisingly, this held true for the majority of neurons. Only few neurons (4% and 12% in TarIn, and 0% and 2% in DisIn, for monkey S and R, respectively) showed opposite results for the two rules (Fig. 6). Thus, our results suggest that the NCE within FEF does not depend on the rules used by the monkeys. Also note that approximately 25% of the neurons showed a higher response in incongruent compared to congruent Tar trials (Fig. 6a, b), which may suggest the existence of different types of conflict processing neurons within FEF.”

8. As discussed in point 3, SNR analysis should be performed computing SNRs relative to neuronal baseline.

As discussed in point 1 and 3, the SNR reflects the target representation and is, in fact, independent of baseline firing. However, if we calculate the SNR relative the baseline, it reflects whether the neuron response is enhanced or suppressed, not the target representation per se. Meanwhile, as discussed in point 3 and Figure 8, the neuronal response decreased as a function of time. Thus, in reality, the “baseline” also changes as a function of time.

Minor:

1. The frontal eye field is strictly speaking at best in the prefrontal cortex although some might even argue that its cellular organization is more alike the premotor cortex than the prefrontal cortex. In any case, the term frontal cortex used in the paper is misleading and I would recommend changing it for prefrontal.

We fully agree with the reviewer's comment and changed the title to “Neuronal congruency effects in macaque prefrontal cortex”.

2. Figure 1. Remove (A) and (B) from legend header as these are inaccurate. 'A' and 'B' are removed.

3. Figure 1. Small box with colored squares and incon/con labels is missing information on task relevance

We added the relevant labels in Figure 1.

4. Figure 1. Please make it clear both in the figure and the text 1) that congruence is defined in terms of match between instruction and RF spatial location and 2) only applies to the rule 1 condition (implicit in figure but should be made explicit)

We explained the congruency in the manuscript -see our reply to the major point 1 of reviewer #1.

5. Please indicate distractor dimming in figure

Distractor dimming is indicated in Figure 2.

6. Page 6, line 19, please cite recent Di Bello et al. (2021, Cerebral Cortical) paper of task-relevant and task irrelevant distractor coding in the FEF.

Thank you for pointing us to this relevant study of Di Bello et al. (2021), which is now cited.

7. Indicate in figure S2 where (what timing) on panel A indexes presented in panels B and C were computed.

The time window is indicated in the legend.

Reviewer #2 (Remarks to the Author):

The study by Yao and Vanduffel on “Neuronal congruency effects in macaque frontal cortex” tackles a very important issue, that is what the neural basis of congruency effects in task switching paradigms such as the Stroop task is. Such tasks that induce interference by means of congruency/incongruency conditions have been widely used in psychology and the study of cognitive control. Yet, surprisingly little is known about their neural basis.

This study has a large number of significant strengths:

1) Probing two versions of the task: The authors designed a novel congruency task that they probed in two versions, a color and a spatial version, demonstrating converging results and thus generalization of their neural results.

2) Behavioral results: The congruency effects are very impressive with this task. My only minor concern is here that performance in the congruent condition is close to ceiling, suggesting that the dimming was quite salient. Was there a particular rationale for having the animals perform close to ceiling?

3) Neural results: The neural results are very clear supporting the authors' hypothesis-driven rationale of target-related increases in signal (i.e. at the attended location) and decreases of noise (at the unattended location) as a function of congruency.

4) Link to behavior: The neural results are also able at least partially to account for

behavior.

This is a very clearly written and presented study on an important topic that reports straightforward results that will add nicely to the literature.

I have only a few comments that the authors may address (in addition to my questions above):

- Temporal structure: The delay period is quite long. Did you mix trials with shorter delay (e.g. <1s) with those that have longer delays (e.g. >1s)?

It would be interesting to look at these trials separately to understand when the spatial location is decided and how well spatial attention is sustained over time. Investigating the 500ms time window before target dimming is a good start, but there could be large differences between short and long trials that influence the main effects that are presented.

- Microsaccades: Is it possible that microsaccades known to modulate neural activity across the attention network contributed to the neural effects? MSs would be executed systematically towards or away from the RF in some conditions and could also influence incongruency effects.

- Rationale for FEF: Please provide a better rationale for FEF. Since FEF neurons do not care about features (just about their spatial implementation), it is perhaps not so surprising that there were no differences between the color and spatial conditions. It is unlikely that the color cue was implemented in FEF. Please discuss how the cueing paradigm may be implemented in FEF, particularly with regard to the results of feature-based attention in PFC reported by the Desimone lab in recent years.

We thank the reviewer for the insightful summary and constructive comments.

Was there a particular rationale for having the animals perform close to ceiling? The main reason to use relatively salient dimming events is because we aimed to obtain performance levels in the incongruent conditions that were not too low (around 80% as shown in Fig.3). On the other hand, there are no reasons to believe that more or less salient dimming events would have a strong effect on the neuronal data, since in this study we were mainly interested in what happens immediately before the dimming.

- Temporal structure: The delay period is quite long. Did you mix trials with shorter delay (e.g. <1s) with those that have longer delays (e.g. >1s)?

Indeed, in our analysis, we mixed trials with short and long delays. Inspired by the reviewer, we now also investigated potential NCE differences between long- and short-delay trials. We found some interesting and surprising results, which are included in Figure 8 and are described in the results section (line 320-339): “The relatively large range of delays between color- and go cue that we used (660-1950ms) provided us with the opportunity to investigate whether the NCE varies as function of delay duration. To this end, we separated short from long delay trials (target dimming occurred respectively 660-1260ms, or 1260-1950ms after color-cue

offset). Note that no target dimming occurred until 660ms after color-cue offset (Fig. 2).

In both monkeys, we found higher SNR in congruent than incongruent conditions for both short (Monkey S: Fig. 8a, median SNR: 1.57 vs 1.19, $p = 5.9e-11$; Monkey R: Fig. 8k, 1.17 vs 1.09, $p = 1.5e-4$) and long delay trials (Monkey S: Fig. 8b, median SNR: 1.50 vs 1.18, $p = 2.4e-9$; Monkey R: Fig. 8l, 1.28 vs 1.01, $p = 2.9e-11$). Since the amplitude of the neuronal response decreases as a function of time after color-cue offset (Fig. S5), the average neuronal response before target dimming is significantly higher in short than long delay trials. This holds for all conditions and both monkeys, regardless of congruency level and irrespective whether a target or distractor is presented within the RF (all $p < 0.0005$, Fig. 8f-i, p-s). Despite this gradual decrease in activity, we did not observe consistent SNR differences between short and long delay trials for both congruent (Fig. 8c, 1.57 vs 1.50, $p = 0.2$) and incongruent (Fig. 8d, 1.19 vs 1.18, $p = 0.084$) conditions in Monkey S. In Monkey R, long delay trials show significantly higher SNRs than short delay trials for congruent conditions (Fig. 8m, 1.28 vs 1.17, $p = 0.041$), yet the opposite effect for incongruent conditions (Fig. 8n, 1.09 vs 1.01, $p = 0.0016$). These results suggest that the SNR in FEF is dominated by the congruency in the current study, and not by the duration of the delay (Fig. 8e, o), nor the amplitude of the neuronal response (Fig. 8j, t). ” and in Fig.8, and in Supplementary Figure 5.

We also modified the abstract (line 23-25) “Such NCE is dominated by the level of congruency, and is not determined by the task rules the subjects used, their reaction times (RT), the length of the delay period, nor the response levels of the neurons,”

In introduction (line 129-131): “Finally, since we used a relatively large range of delays between color-cue and target dimming (660-1950 ms, Fig. 2), we are also able to investigate the effect of delay duration on the NCE.”

and in discussion (line 413-422) “Moreover, the large range of delays between the color-cue and target dimming (go cue) enabled us to investigate the effect of delay duration on the NCE. We found a significant NCE for both long and short delay trials, which is consistent with a human behavioral study using a delayed match-to-sample Stroop task⁶⁰. Surprisingly, however, we did not observe consistent differences in NCE between short and long delay trials (Fig. 8). Moreover, the neuronal response is significantly higher in short versus long delay trials (Fig. 8). These results suggest that the NCE as indexed by the SNR before target dimming is dominated by the level of congruency, and that it is not determined by the RT (Fig. 7m, n), the length of the delay (Fig. 8e, o), and the responsiveness of the neurons (Fig 8j, t). Yet, it may be sensitive to other cognitive processes that affect RT (Supplementary Fig. 3).”

Microsaccades: Is it possible that microsaccades known to modulate neural activity across the attention network contributed to the neural effects? MSs would be executed systematically towards or away from the RF in some conditions and could also influence incongruency effects.

Microsaccades may be a possible behavioral indicator of the congruency effect, and studying microsaccades may help increase our understanding of the NCE and the relationship between NCE and attention (Lowet, et al 2018, Neuron). We mentioned this possibility and we think it is an interesting future research direction (line 442-444). “Where the NCE emerges in the brain, what the relationship is with selective attention, and other related behaviors (such as microsaccades)⁶³, however, remains to be investigated in the future studies.”

Unfortunately, the sampling rate of our eye tracking system is only 120Hz. Therefore, we wish to refrain from conclusive analyses in this respect. For example, based on a paper investigating the relationship of attention and microsaccades, the threshold for the minimum duration of the microsaccade was set to 8 ms (Lowet, et al 2018, Neuron), which would be exceedingly challenging, not to say impossible, to achieve with the current data.

- Rationale for FEF: Please provide a better rationale for FEF. Since FEF neurons do not care about features (just about their spatial implementation), it is perhaps not so surprising that there were no differences between the color and spatial conditions. It is unlikely that the color cue was implemented in FEF. Please discuss how the cueing paradigm may be implemented in FEF, particularly with regard to the results of feature-based attention in PFC reported by the Desimone lab in recent years.

We now detailed the rationale for targeting the FEF in the introduction (line 60-68) and also cited the relevant Desimone paper (Bichot, et al 2019, nature communications): “The FEF contains a retinotopic map of visual saliency, as it integrates the bottom-up driven intrinsic saliency of visual stimuli with top-down signals (e.g., attention, experience, reward expectation, goals, knowledge etc.). The peak activity within the saliency map indicates the purported target location in the visual field for further processing³⁴. Therefore, if a task-irrelevant feature in an incongruent condition is processed to some degree, it may affect the saliency map through spatial or feature-based processes generated within the FEF, or fed to the FEF from neighboring areas within dorsal lateral prefrontal cortex³⁵. Thus, the FEF is an ideal area to investigate differences in target representation between congruent and incongruent conditions, independent whether the spatial location or a feature (color) of a cue is used³⁶”.

Figure RR1 (corresponding to the reviewer #1, major point #3). The distribution of the visual modulation index of all neurons for each condition from two monkeys. As expected, almost all neurons showed positive visual modulation index in each conditions, which confirmed that the neurons responded higher after the visual stimuli onset, i.e., the neurons are visual excitatory neurons. The visual modulation index was calculated as the difference between the visual response (500 ms interval immediately after stimuli onset) and the baseline response (400 ms interval prior the task-rule cue) divided by the sum of the two.

Figure RR2 (corresponding to the reviewer #1, major point #5). The distribution of the reaction times for each condition. The trials are pooled across sessions. The distributions of raw RTs argue against express RTs.

REVIEWERS' COMMENTS

Reviewer #1 (Remarks to the Author):

I commend the authors for their revision. At this time, all my concerns have been addressed and I do not have any further question regarding this manuscript. I thus now recommend its publication as is. Congratulations to the authors on this work.

Reviewer #2 (Remarks to the Author):

The authors have done a thorough job with the revision, and I have no further comments. Congratulations on an interesting contribution to our field!

REVIEWERS' COMMENTS

Reviewer #1 (Remarks to the Author):

I commend the authors for their revision. At this time, all my concerns have been addressed and I do not have any further question regarding this manuscript. I thus now recommend its publication as is. Congratulations to the authors on this work.

Reviewer #2 (Remarks to the Author):

The authors have done a thorough job with the revision, and I have no further comments. Congratulations on an interesting contribution to our field!

We appreciate the reviewers for their positive criticism and insightful suggestions, which leads to the publication of this manuscript.

We made few changes on the following figures:

1, we changed the layout of the figure 6 to make it more consistent with other figures. We modulated its legends and the main text accordingly.

2, We changed the color of the dots in figure 7 and 8 to gray.